# Facilitators and barriers of infectious diseases surveillance activities: lessons from the Global Polio Eradication Initiative - a mixed-methods study

Ikponmwosa Osaghae [1,2] Priyanka Agrawal [1] Adetoun Olateju,[1] Olakunle Alonge[1]

[1]Department of International Health, Johns Hopkins Bloomberg School of Public Health, Baltimore, Maryland, USA
[2]Department of Epidemiology, Human Genetics and Environmental Sciences, The University of Texas Health Science Center at Houston, Houston, Texas, USA

**Correspondence to**
Dr Olakunle Alonge;
oalonge1@jhu.edu

## ABSTRACT

**Objectives** To document lessons from the Global Polio Eradication Initiative (GPEI) by determining factors associated with successful surveillance programme globally as well as at national and subnational levels. The process of conducting surveillance has been previously recognised in the literature as important for the success of polio surveillance activities.

**Design** A cross-sectional survey with closed and open-ended questions.

**Settings** Survey of persons involved in the implementation of surveillance activities under the GPEI at the global level and in seven low-income and middle-income countries.

**Participants** Individuals (n=802) with ≥12 months of experience implementing surveillance objective of the GPEI between 1988 and 2019.

**Main outcome measures and methods** Quantitative and qualitative analyses were conducted. Logistic regression analyses were used to assess factors associated with implementation process as a factor for successful surveillance programme. Horizontal analysis was used to analyse qualitative free-text responses on facilitators and barriers identified for conducting surveillance activities successfully.

**Results** Overall, participants who reported challenges relating to GPEI programme characteristics had 50% lower odds of reporting implementation process as a factor for successful surveillance (adjusted OR (AOR): 0.50, 95% CI: 0.29 to 0.85). Challenges were mainly perceptions of external intervention source (ie, surveillance perceived as 'foreign' to local communities) and the complexity of surveillance processes (ie, surveillance required several intricate steps). Those who reported organisational challenges were almost two times more likely to report implementation process as a factor for successful surveillance (AOR: 1.89, 95% CI: 1.07 to 3.31) overall, and over threefolds (AOR: 3.32, 95% CI: 1.14 to 9.66) at the national level.

**Conclusions** Programme characteristics may have impeded the process of conducting surveillance under the GPEI, while organisational characteristics may have facilitated the process. Future surveillance programmes should be designed with inputs from local communities and frontline implementers.

## STRENGTHS AND LIMITATIONS OF THIS STUDY

⇒ This study reports challenges and factors associated with successful polio surveillance from the perspectives of implementers directly involved in polio surveillance since the launch of the Global Polio Eradication Initiative.
⇒ Data for this study were collected at the global and national levels increasing the potential for generalisability across countries and settings.
⇒ Since participants reported their experiences over some time in the past, the study may be prone to recall bias.
⇒ Given the cross-sectional study design, we are unable to infer causality from this study.

## INTRODUCTION

The Global Polio Eradication Initiative (GPEI) was launched in 1988 with the goal of attaining a polio-free world. Since its launch, over 200 countries have been involved in this initiative with national governments leading its implementation.[1] Currently, polio remains endemic in two countries of the WHO Eastern Mediterranean Region—Afghanistan and Pakistan.[2] Over the last three decades of implementing the GPEI, surveillance has remained one of the key strategies for interrupting the transmission of poliovirus and tracking progress.[3–5] Acute flaccid paralysis (AFP) and environmental surveillance remain gold standards for the detection of polio cases, especially in endemic and high-risk countries.[6 7] While the attainment of a polio-free world may appear delayed, much has been achieved since the initiative kicked off in 1988. Following the eradication of wild poliovirus type 2 in 2015, and wild poliovirus type 3 more recently, the global polio community is a step closer to the finish line, with concerted efforts at eradication of wild poliovirus type 1 currently underway.[8]

Disease surveillance is a key public health strategy that has played a vital role in improving our understanding of epidemic trends and developing sound strategies for disease prevention and control, including tuberculosis, HIV, SARS, Ebola and more recently, SARS-CoV-2.[9–13] The ongoing SARS-CoV-2 pandemic—with over 500 million infections and over 6 million deaths globally as of May 2022—has further highlighted the need for more robust global health system surveillance strategy.[14 15] Evidently, the implementation of activities to achieve the objective of surveillance as part of the GPEI has experienced successes and challenges in different regions globally which portends important lessons for other disease control activities, especially in similar settings and contexts.[16]

Various factors including characteristics of individuals involved in surveillance, their organisational settings, the characteristics of the GPEI programme and its implementation process, and other external factors have played a role in shaping surveillance activities under the polio programme at global and local levels.[17] The process of conducting surveillance is especially critical for the successful implementation of the GPEI.[18] Implementation process which is often continuous with several inter-related and compensatory components is a major determining factor for the successful implementation of any intervention, and ranges from planning, engagement, execution to evaluation.[18–21] Individual characteristics (eg, knowledge and perception) of the implementers can be enhanced through training and retraining—and such capacity-building activities could improve the implementation process and successful implementation of disease surveillance.[22 23] Hamisu *et al* found that in security-challenged Nigerian states, annual training of district surveillance officers and monthly progress meetings were among implementation strategies used to achieve successful surveillance through improvements in the implementation process.[24]

Although polio surveillance has been previously described both at the global and national level, few studies have systematically documented lessons from the perspective of participants directly involved in polio surveillance activities.[25] We aim to document lessons from the GPEI by determining factors associated with a successful polio surveillance programme. Polio surveillance anchors all disease surveillance activities in most low-income and middle-income countries (LMICs). Hence, such lessons are important in understanding how various contextual factors may influence the implementation of surveillance activities more broadly in LMICs. We expect that the findings will provide crucial insights toward the final eradication of polio and help in addressing current and future disease epidemics.

## METHODS

### Data source and study population

This cross-sectional study used data from the Synthesis and Translation of Innovation and Research from Polio Eradication project which aimed to apply implementation science methods to document lessons learnt from GPEI by capturing perspectives across the polio universe on various GPEI strategies including surveillance.[17] The survey tool, published elsewhere,[17] was developed based on the Consolidated Framework for Implementation Research to describe programme, organisational and implementer characteristics that influenced the process of implementing various GPEI objectives.[19] The survey with close and open-ended questions was administered online through Qualtrics, a web-based data collection platform. The polio universe consisted of individuals who were directly involved in implementing surveillance activities under the GPEI at the global level including seven LMICs (Afghanistan, Bangladesh, the Democratic Republic of Congo, Ethiopia, India, Indonesia and Nigeria).[26] The study population included funders, managers, policymakers, researchers and frontline field implementers who have spent 12 or more continuous months working on surveillance activities under the GPEI between 1988 and 2019.[26] We explored the influence of various factors on the successful implementation of the polio surveillance programme (online supplemental table 1 and online supplemental figure 1). Embedded mixed-method approach was used to collect both quantitative and qualitative data in the survey.[27]

The study protocol conforms to the ethical guidelines of the Declaration of Helsinki.

## Measures

### Dependent variable

The dependent variable was defined as 'internal factors for a successful surveillance program'. Respondents were asked, 'In your opinion, which of the following was the biggest internal contributor to your program's success in completing its objective of surveillance?' possible responses (ie, internal factors) included the implementation process (eg, planning, engaging), individual attributes (eg, knowledge, self-efficacy), organisational characteristics (eg, organisation support for surveillance activities) and polio programme characteristics (eg, technologies used as part of the programme) (see online supplemental table 1). Success was defined as the respondent's perceived accomplishment of the primary objective for implementing surveillance activities (eg, whether the objective to detect AFP cases was achieved or not). Since the implementation process is closely linked to the individual characteristics of implementers, we combined these two constructs. Those who reported both implementation process and individual characteristics as the biggest internal factor for a successful surveillance programme were categorised to represent implementation process as a factor for successful surveillance programme, while all those who did not report either process of conducting activities and individual characteristics were classified as otherwise (the implementation process is not a factor for successful surveillance programme). Hence, the final dependent variable was categorised as a binary variable

(implementation process as a factor for successful surveillance programme, Yes or No) (see online supplemental figure 1).

## Independent variables

The main explanatory variables were the three main challenges (barriers) in the implementation of surveillance activities, operationalised as a binary variable (yes/no). Challenges were: (1) Organisational challenges—factors relating to characteristics of the organisation supporting the polio eradication programme, (2) GPEI programme challenges—defined as challenges relating to specific activities used towards eradicating polio and (3) External challenges—defined as challenges relating to the impact of political, economic, social, technological or environmental settings (detailed definition are in online supplemental table 1 and online supplemental figure 1). Other explanatory variables were the respondents' characteristics including, their level of involvement (global, national and subnational levels), region of surveillance (Africa, Southeast Asia, Eastern Mediterranean and Western Hemisphere), years of experience (<4 years, 5–9 years, 10–14 years, 15–19 years and >20 years), primary role (advisory, management, supervisory, frontline and other) and country status of polio endemicity (endemic (Nigeria, Afghanistan and Pakistan) vs non-endemic (countries certified as polio-free)). We defined polio endemicity based on the polio status of country as at when participants completed the survey,[2] which was before Nigeria was declared polio-free. A total of 70 countries represented in the survey were included in the polio surveillance analyses. However, the sample size of respondents from individual countries were limited to allow country-specific analyses. Qualitative data were collected in the survey as short, free-text responses which provided a contextual understanding of the quantitative measures.

## Data analysis

The data set was restricted at two levels, (1) implementation of surveillance objective between 1988 and 2019, and (2) implementation at national and district levels, to highlight recent lessons on surveillance from the frontlines of polio eradication. The analytical data set contained individuals who have been involved in surveillance activities between 1988 and 2019 for at least 12 continuous months. Of a total of 3659 individuals who completed the survey, 1624 (44.38%) individuals had been involved with polio activities prior to 1988, and 68 (0.02) individuals had less than 12 months of experience with GPEI activities and thus were dropped from the analysis as per the inclusion criteria. Of the remaining 1967 (53.76%) individuals who fit the inclusion criteria, 802 (40.77%) individuals were involved in different capacities with the objective of surveillance and were the final analytical sample for this paper (online supplemental figure 1). We described the distribution of surveillance challenges and other explanatory variables stratified by the dependent variable (implementation process or otherwise) using

simple proportion and $\chi^2$ test. Complete case analyses were conducted. Bivariable and multivariable logistics regression models were used to assess the odds of implementers reporting process of implementation as a factor for successful surveillance programme compared with not reporting process of implementation. Purposeful and forward stepwise model selection was used to select variables and build best-fit regression models. Akaike information criterion (AIC), was then used to evaluate model fitness. The best fit model (AIC=783.7) retained polio programme challenges, organisational challenges, years of experience and region of surveillance. However, external challenges, level of involvement, primary role and country polio status were considered relevant to the study aim and were also included in our final multivariable regression models. Subgroup analyses were performed at the national and district level as well as specifically for the African region (given over-representation of respondents from the African region). All quantitative analyses were conducted in Stata/IC V.15.1.[28] Statistical significance was defined as a two-sided p value<0.05 for all comparisons. Furthermore, horizontal analysis was done to systematically identify commonalities among respondents on the key challenges and factors for successful polio surveillance programmes across the different regions.[29]

## Patient and public involvement

There were no patients and/or public involvement in the study design, setting of research questions, study implementation or dissemination plans of this research.

## RESULTS

A total of 802 participants who provided responses on internal factors for successful polio surveillance programme were included in this study. A total of 511 (63.72%) participants reported the process of implementing surveillance as a factor for successful surveillance and 291 (36.28%) reported otherwise. According to a subnational surveillance officer from our qualitative analysis, process activities identified as factors for successful surveillance included 'the setting up of community-based surveillance which involved community informants within their cells or villages, health workers and effective collaboration between community teams and district officials in disease detection, reporting, investigation, feedback and outbreak response'. A district polio programme supervisor described the 'expansion of surveillance focal sites and adoption of new strategies like environmental surveillance' within communities.

Regular training, financial incentives and routine progress meetings were additional internal contributors to successful surveillance described by respondents. For example, a frontline polio vaccinator observed that 'incentives were given to informants on any AFP reported, financial reimbursements were provided to support surveillance training and meetings;' while a polio project lead at an implementing non-governmental organisation

remarked 'competency of staff was a priority, so surveillance focal point persons benefitted from high-quality capacity-building programs;' and a national surveillance officer recalled that 'we had regular meetings with surveillance country teams to monitor and evaluate progress, assess performance and plan new approaches together'.

Furthermore, the incorporation of technology tools facilitated reporting and could have contributed to quicker detection of outbreaks. According to a national surveillance officer, 'the use of modern technology in data transfer and stool transport, including setting up of electronic surveillance gave the advantage of being instantly tracked on a server with very quick response time and timely support'.

Participants who reported GPEI programme challenges were less likely to report implementation as an internal factor for successful surveillance (p value=0.031). Also, we found a significant difference in the years of experience with surveillance among those who reported implementation process as a factor for successful surveillance compared with those who reported otherwise (p value=0.048), with fewer of those who have had 10–14 years of experience with surveillance reporting implementation process as a contributor to surveillance success (table 1).

However, there was no significant difference in report of organisational challenges and external challenges among those who reported implementation process as an internal factor for successful surveillance compared with those who reported otherwise. Similarly, we found no significant difference in the distribution of level of surveillance involvement, and region of surveillance in participants who reported implementation process as a factor for successful surveillance compared with those who reported otherwise. Among participants who reported implementation process as a factor for successful surveillance, 45.6% worked in a frontline role, 21.4% worked in a supervisory role, 4.9% worked in a management role and 2.7% worked in an advisory role. However, there was no significant difference in the distribution of those who reported implementation process compared with those who reported otherwise. Although the majority of participants in our study worked in non-endemic countries, polio endemicity status of countries had no significant difference in the distribution of participants who reported implementation process compared with those who reported otherwise (table 1).

On further disaggregation, we found that those reporting implementation process as a main factor for success were more likely to report external challenges such as global climate and ineffective cross-organisational collaboration (p=0.007) (online supplemental table 2).

Overall, participants who reported GPEI programme challenges had 50% (adjusted OR (AOR): 0.50, 95% CI: 0.29 to 0.85) significantly lower odds of reporting implementation process as a factor for successful surveillance programme compared with those who did not report GPEI programme challenges (table 2).

We found the main GPEI programme challenge reported were related to the intervention source and complexity of implementation. There was a perception that the source of the surveillance interventions was external to the benefitting communities which made the surveillance process for polio a challenge during implementation. A district surveillance officer remarked that 'parents were sometimes not supportive, some thought polio was not their priority, they were skeptical to release their child's faecal samples, so we not only had to continue educating them but also gave out small incentives to encourage them to participate'. Complexity of implementation included the perception that the surveillance process included several intricate steps, was of long duration, required special handling and transportation. A supervising surveillance officer observed that '"when AFP cases were reported from community networks, there were delays in the transportation of samples and confirmatory feedback from the district level or national labs took several days, increasing the risk of an outbreak'.

On the other hand, those who reported organisational challenges were almost two times (AOR: 1.89, 95% CI: 1.07 to 3.31) significantly more likely to report implementation process as a factor for successful surveillance programme compared with those who did not report the same. Also, participants who had 10–14 years of experience had 44% (AOR: 0.56, 95% CI: 0.33 to 0.94) significantly lower odds of reporting implementation process as a factor for successful surveillance programme compared with those with <5 years of experience (table 2).

At the national level, we found that those who reported organisational challenges were over threefolds (AOR: 3.32, 95% CI: 1.14 to 9.66) significantly more likely to report implementation process as a factor for successful surveillance programme compared with those who did not report the same. Also, those who had 15–19 years of experience had 79% (AOR: 0.21, 95% CI: 0.06 to 0.66) significantly lower odds of reporting implementation process as a factor for successful surveillance programme compared with those with <5 years of experience (table 3).

At the district level, reporting implementation process as a factor for successful surveillance was not associated with reporting GPEI programme challenges, organisational challenges, external challenges, region, years of surveillance experience, role or country polio status in both bivariable and multivariable analyses (table 4).

Participants in the African region who reported GPEI programme challenges had 62% (AOR: 0.38, 95% CI: 0.19 to 0.75) significantly lower odds of reporting implementation process as a factor for successful surveillance programme compared with those who did not report GPEI programme challenges. Also, participants from the African region with 5–9 years experience or 10–14 years experience in surveillance activities had 47% and 57% significantly lower odds respectively of reporting implementation process as a factor for successful surveillance compared with those with <5 years experience (table 5).

**Table 1** Distribution of surveillance challenges and relevant characteristics stratified by internal factors for successful surveillance

| Challenges/surveillance characteristics | Internal factors for successful surveillance | | P value |
|---|---|---|---|
| | Implementation process (N=511) | Others (N=291) | |
| Organisational challenges, n (%) | | | 0.124 |
| Yes | 87 (17.30) | 37 (13.12) | |
| No | 416 (82.70) | 245 (86.88) | |
| GPEI programme challenges, n (%) | | | 0.031 |
| Yes | 52 (10.34) | 44 (15.60) | |
| No | 451 (89.66) | 238 (84.40) | |
| External challenges*, n (%) | | | 0.949 |
| Yes | 261 (51.89) | 147 (52.13) | |
| No | 242 (48.11) | 135 (47.87) | |
| Level of involvement of surveillance†, n (%) | | | 0.343 |
| Global level | 22 (4.31) | 7 (2.41) | |
| National level | 35 (6.85) | 23 (7.93) | |
| Subnational level | 454 (88.85) | 260 (89.66) | |
| Region of surveillance‡, n (%) | | | 0.269 |
| Africa | 219 (54.34) | 129 (57.33) | |
| Southeast Asia | 79 (19.60) | 32 (14.22) | |
| Eastern Mediterranean | 87 (21.59) | 49 (21.78) | |
| Western Hemisphere | 18 (4.47) | 15 (6.67) | |
| Years of experience with surveillance†, n (%) | | | 0.048 |
| <5 years | 132 (25.88) | 65 (22.34) | |
| 5–9 years | 138 (27.06) | 82 (28.18) | |
| 10–14 years | 98 (19.22) | 74 (25.43) | |
| 15–19 years | 78 (15.29) | 49 (16.84) | |
| >20 years | 64 (12.55) | 21 (7.22) | |
| Primary role†, n (%) | | | 0.627 |
| Advisory | 14 (2.74) | 6 (2.07) | |
| Management | 25 (4.89) | 8 (2.76) | |
| Supervisory | 108 (21.14) | 62 (21.38) | |
| Frontline | 233 (45.60) | 135 (46.55) | |
| Other | 131 (25.64) | 79 (27.24) | |
| Country polio status, n (%) | | | 0.939 |
| Non-endemic | 371 (72.60) | 212 (72.85) | |
| Endemic | 140 (27.40) | 79 (27.15) | |

*17 (2.12%) observations missing.
†1 (0.01%) observations missing.
‡174 (21.70%) observations missing.
GPEI, Global Polio Eradication Initiative.

## DISCUSSION

From our study, two-thirds of implementers involved in polio surveillance activities as part of the GPEI efforts from 1988 to 2019 reported implementation process as a factor for surveillance success. Our findings support a previous report that GPEI planning processes with a multiyear strategic plan are critical for improving surveillance.[18] Successful surveillance depends on detailed planning of the entire process necessary for the execution of surveillance activities including the engagement of individuals with the right knowledge, perception and self-efficacy. Regions where poliovirus transmission has persisted, have been linked with poorly established processes of active surveillance and response to polio cases.[30–33]

However, our study found no difference in surveillance implementation process in countries where polio remains endemic compared with those where polio has been eradicated. Additionally, we found that role within the GPEI had no influence on the implementation process as a factor for surveillance success. These findings suggest

**Table 2** Bivariable and multivariable logistic regression analyses showing factors associated with implementation process as a factor for successful surveillance programme in overall study population

| Surveillance characteristics | Unadjusted | | Adjusted | |
|---|---|---|---|---|
| | OR (95% CI) | P value | OR* (95% CI) | P value |
| GPEI programme challenges | | | | |
| No | Ref | Ref | Ref | Ref |
| Yes | 0.62 (0.41 to 0.96) | 0.032 | 0.50 (0.29 to 0.85) | 0.01 |
| Organisational challenges | | | | |
| No | Ref | Ref | Ref | Ref |
| Yes | 1.38 (0.91 to 2.10) | 0.125 | 1.89 (1.07 to 3.31) | 0.027 |
| External challenges | | | | |
| No | Ref | Ref | Ref | Ref |
| Yes | 1.02 (0.77 to 1.36) | 0.879 | 0.88 (0.61 to 1.25) | 0.471 |
| Level of involvement of surveillance | | | | |
| Global level | Ref | Ref | Ref | Ref |
| National level | 0.48 (0.18 to 1.32) | 0.155 | 0.62 (0.19 to 2.01) | 0.424 |
| Subnational level | 0.56 (0.23 to 1.32) | 0.182 | 0.70 (0.25 to 1.91) | 0.483 |
| Region of surveillance | | | | |
| Africa | Ref | Ref | Ref | Ref |
| Southeast Asia | 1.45 (0.91 to 2.31) | 0.114 | 1.42 (0.85 to 2.37) | 0.184 |
| Eastern Mediterranean | 1.05 (0.69 to 1.58) | 0.831 | 1.01 (0.61 to 1.66) | 0.981 |
| Western Hemisphere | 0.71 (0.34 to 1.45) | 0.344 | 0.70 (0.32 to 1.54) | 0.378 |
| Years of experience with surveillance | | | | |
| <5 years | Ref | Ref | Ref | Ref |
| 5–9 years | 0.83 (0.55 to 1.24) | 0.362 | 0.66 (0.41 to 1.06) | 0.086 |
| 10–14 years | 0.65 (0.43 to 1.00) | 0.048 | 0.56 (0.33 to 0.94) | 0.028 |
| 15–19 years | 0.78 (0.49 to 1.25) | 0.304 | 0.75 (0.42 to 1.32) | 0.315 |
| >20 years | 1.50 (0.84 to 2.67) | 0.167 | 1.99 (0.92 to 4.32) | 0.081 |
| Primary role, n (%) | | | | |
| Advisory | Ref | Ref | Ref | Ref |
| Management | 1.34 (0.39 to 4.65) | 0.645 | 1.19 (0.25 to 5.74) | 0.832 |
| Supervisory | 0.75 (0.27 to 2.04) | 0.569 | 0.63 (0.18 to 2.23) | 0.475 |
| Frontline | 0.74 (0.28 to 1.97) | 0.546 | 0.66 (0.19 to 2.29) | 0.518 |
| Other | 0.71 (0.26 to 1.92) | 0.502 | 0.63 (0.18 to 2.22) | 0.473 |
| Country polio status, n (%) | | | | |
| Non-endemic | Ref | Ref | Ref | Ref |
| Endemic | 1.01 (0.73 to 1.40) | 0.939 | 1.11 (0.72 to 1.72) | 0.635 |

*OR from the logistic regression model was adjusted for GPEI programme characteristics, organisational characteristics, external settings, level of involvement, region, years of surveillance experience, primary role and country polio status.
GPEI, Global Polio Eradication Initiative; Ref, Reference.

that factors other than implementation process may be necessary to achieve the polio surveillance objective in endemic countries. For example, prior to polio elimination in Nigeria, the building of surveillance infrastructure and technology such as the Geographical Information System were reported as key lessons that facilitated polio surveillance programme.[34]

From our qualitative analysis, we found that the expansion of surveillance focal sites beyond health facilities to include community surveillance was one of the factors for successful surveillance. Community-based polio surveillance programmes in Ethiopia which focused on AFP detection among remote and migratory populations (Core Group Polio Project) contributed to increasing

**Table 3** Bivariable and multivariable logistic regression analyses showing factors associated with implementation process as a factor for successful surveillance programme at the national level

| Surveillance characteristics | Unadjusted | | Adjusted | |
|---|---|---|---|---|
| | OR (95% CI) | P value | OR* (95% CI) | P value |
| GPEI programme challenges | | | | |
| No | Ref | Ref | Ref | Ref |
| Yes | 0.88 (0.44 to 1.78) | 0.73 | 0.63 (0.23 to 1.71) | 0.365 |
| Organisational challenges | | | | |
| No | Ref | Ref | Ref | Ref |
| Yes | 1.41 (0.76 to 2.61) | 0.275 | 3.32 (1.14 to 9.66) | 0.028 |
| External challenges | | | | |
| No | Ref | Ref | Ref | Ref |
| Yes | 0.84 (0.51 to 1.36) | 0.469 | 0.54 (0.26 to 1.12) | 0.115 |
| Region of surveillance | | | | |
| Africa | Ref | Ref | Ref | Ref |
| Southeast Asia | 1.00 (0.36 to 2.78) | 1 | 1.61 (0.47 to 5.55) | 0.449 |
| Eastern Mediterranean | 0.84 (0.39 to 1.84) | 0.668 | 1.21 (0.48 to 3.07) | 0.687 |
| Western Hemisphere | 0.63 (0.26 to 1.53) | 0.302 | 0.69 (0.25 to 1.86) | 0.46 |
| Years of experience with surveillance | | | | |
| <5 years | Ref | Ref | Ref | Ref |
| 5–9 years | 1.03 (0.56 to 1.90) | 0.929 | 0.61 (0.20 to 1.83) | 0.375 |
| 10–14 years | 0.71 (0.37 to 1.37) | 0.309 | 0.38 (0.13 to 1.11) | 0.077 |
| 15–19 years | 0.97 (0.48 to 1.92) | 0.92 | 0.21 (0.06 to 0.66) | 0.007 |
| >20 years | 2.33 (0.99 to 5.52) | 0.054 | 0.80 (0.21 to 3.06) | 0.748 |
| Primary role, n (%) | | | | |
| Advisory | Ref | Ref | Ref | Ref |
| Management | 0.96 (0.16 to 5.90) | 0.965 | 2.00 (0.19 to 21.34) | 0.566 |
| Supervisory | 1.09 (0.23 to 5.15) | 0.909 | 2.32 (0.35 to 15.28) | 0.38 |
| Frontline | 0.97 (0.22 to 4.23) | 0.965 | 2.13 (0.36 to 12.72) | 0.406 |
| Other | 1.12 (0.25 to 5.01) | 0.885 | 2.48 (0.40 to 15.52) | 0.332 |
| Country polio status, n (%) | | | | |
| Non-endemic | Ref | Ref | Ref | Ref |
| Endemic | 1.17 (0.64 to 2.15) | 0.605 | 1.32 (0.58 to 3.00) | 0.507 |

*OR from the logistic regression model was adjusted for GPEI programme characteristics, organisational characteristics, external settings, region, years of surveillance experience, primary role and country polio status.
GPEI, Global Polio Eradication Initiative; Ref, Reference.

the non-polio AFP detection nationally, an indication of effective polio surveillance.[35] GPEI over time has developed effective surveillance and monitoring systems that have been expanded to include other vaccine-preventable diseases such as measles and rubella at the community levels with linkages across programmes.[18 35 36] Systematic gaps in surveillance systems, such as insecurity and lack of access to populations that promote transmission were noted in Nigeria, Afghanistan and Pakistan, slowing down surveillance efforts and control of poliovirus.[30 37] In Spain, the maintenance of an effective surveillance system for acute flaccid paralysis was vital for the successes achieved.[31]

Challenges related to GPEI programme characteristics were associated with reduced odds of reporting implementation process as a factor for surveillance success— which can be interpreted to mean that challenges with programme characteristics negatively impact surveillance success. This finding held true in our subset analysis conducted in the African region which was certified free of polio in 2020.[2] Factors such as the complexity and cost of a programme have a huge impact on the success of a surveillance programme. The complexity of polio surveillance including completeness and timeliness of AFP reporting, stool adequacy, sample transportation and processing, high sensitivity and continued

**Table 4** Bivariable and multivariable logistic regression analyses showing factors associated with implementation process as a factor for successful surveillance programme at the district level

| Surveillance characteristics | Unadjusted | | Adjusted | |
|---|---|---|---|---|
| | OR (95% CI) | P value | OR* (95% CI) | P value |
| GPEI programme challenges | | | | |
| No | Ref | Ref | Ref | Ref |
| Yes | 0.77 (0.46 to 1.29) | 0.322 | 0.69 (0.36 to 1.33) | 0.269 |
| Organisational challenges | | | | |
| No | Ref | Ref | Ref | Ref |
| Yes | 1.03 (0.63 to 1.69) | 0.895 | 1.11 (0.56 to 2.21) | 0.756 |
| External challenges | | | | |
| No | Ref | Ref | Ref | Ref |
| Yes | 0.84 (0.51 to 1.36) | 0.469 | 0.87 (0.55 to 1.37) | 0.542 |
| Region of surveillance | | | | |
| Africa | Ref | Ref | Ref | Ref |
| Southeast Asia | 1.40 (0.79 to 2.50) | 0.248 | 1.30 (0.69 to 2.46) | 0.417 |
| Eastern Mediterranean | 0.68 (0.39 to 1.20) | 0.188 | 0.70 (0.37 to 1.33) | 0.281 |
| Western Hemisphere | 0.61 (0.27 to 1.41) | 0.251 | 0.61 (0.25 to 1.49) | 0.274 |
| Years of experience with surveillance | | | | |
| <5 years | Ref | Ref | Ref | Ref |
| 5–9 years | 1.13 (0.68 to 1.89) | 0.631 | 0.91 (0.49 to 1.68) | 0.766 |
| 10–14 years | 0.72 (0.43 to 1.23) | 0.23 | 0.56 (0.29 to 1.08) | 0.084 |
| 15–19 years | 1.03 (0.58 to 1.85) | 0.914 | 1.14 (0.54 to 2.40) | 0.729 |
| >20 years | 1.66 (0.80 to 3.43) | 0.171 | 2.83 (0.93 to 8.61) | 0.067 |
| Primary role, n (%) | | | | |
| Advisory | Ref | Ref | Ref | Ref |
| Management | 1.19 (0.26 to 5.34) | 0.825 | 0.54 (0.08 to 3.80) | 0.533 |
| Supervisory | 0.85 (0.25 to 2.96) | 0.802 | 0.47 (0.09 to 2.54) | 0.378 |
| Frontline | 0.70 (0.21 to 2.32) | 0.556 | 0.46 (0.09 to 2.42) | 0.36 |
| Other | 0.66 (0.19 to 2.25) | 0.505 | 0.46 (0.08 to 2.51) | 0.368 |
| Country polio status, n (%) | | | | |
| Non-endemic | Ref | Ref | Ref | Ref |
| Endemic | 0.85 (0.56 to 1.29) | 0.436 | 1.04 (0.60 to 1.80) | 0.901 |

*OR from the logistic regression model was adjusted for GPEI programme characteristics, organisational characteristics, external settings, region, years of surveillance experience, primary role and country polio status.
GPEI, Global Polio Eradication Initiative; REF, Reference.

quality assurance at testing laboratories have been documented.[38] Maya and colleagues noted a lack of specificity of funding for polio personnel and assets in all 10 countries examined in their study with substantial differences in the detail, accuracy and feasibility of costs in expanded programme on immunisation plans.[39] This underscores the importance of detailed, collaborative and systematic costing for polio programmes to improve the quality of costing and comparability within and across countries. However, this process should account for the peculiarity of each region/level through the engagement of community leaders and relevant stakeholders. This will enhance the quality of study design, identification of context-specific challenges, development of tailored programmes and ultimately promotion of programme adoption and success.[22]

Conversely, challenges related to the organisational characteristics, especially at the national level, were associated with increased odds of reporting process of surveillance as the most important contributor to surveillance success—and did not negatively impact surveillance success. Importantly, the process for implementation of surveillance systems should take into cognizance the peculiarity that exists in different settings and countries. For example, in certain parts of Nigeria, Iraq, Yemen, Syria, and areas that contend with insecurity, accessibility

**Table 5** Bivariable and multivariable logistic regression analyses showing factors associated with implementation process as a factor for successful surveillance programme in Africa

| Surveillance characteristics | Unadjusted | | Adjusted | |
| --- | --- | --- | --- | --- |
| | OR (95% CI) | P value | OR* (95% CI) | P value |
| GPEI programme challenges | | | | |
| No | Ref | Ref | Ref | Ref |
| Yes | 0.46 (0.25 to 0.87) | 0.017 | 0.38 (0.19 to 0.75) | 0.005 |
| Organisational challenges | | | | |
| No | Ref | Ref | Ref | Ref |
| Yes | 1.07 (0.56 to 2.06) | 0.83 | 1.27 (0.63 to 2.57) | 0.507 |
| External challenges | | | | |
| No | Ref | Ref | Ref | Ref |
| Yes | 1.21 (0.78 to 1.87) | 0.397 | 0.96 (0.60 to 1.53) | 0.861 |
| Years of experience with surveillance | | | | |
| <5 years | Ref | Ref | Ref | Ref |
| 5–9 years | 0.59 (0.34 to 1.05) | 0.073 | 0.53 (0.29 to 0.98) | 0.042 |
| 10–14 years | 0.48 (0.25 to 0.92) | 0.028 | 0.43 (0.21 to 0.87) | 0.019 |
| 15–19 years | 0.94 (0.44 to 1.99) | 0.871 | 0.90 (0.40 to 2.01) | 0.797 |
| >20 years | 3.64 (1.02 to 13.02) | 0.047 | 3.66 (0.98 to 13.69) | 0.053 |
| Primary role, n (%) | | | | |
| Advisory | Ref | Ref | Ref | Ref |
| Management | 1.13 (0.18 to 6.93) | 0.899 | 1.09 (0.17 to 7.16) | 0.925 |
| Supervisory | 0.76 (0.18 to 3.19) | 0.703 | 1.00 (0.22 to 4.51) | 0.996 |
| Frontline | 0.88 (0.21 to 3.64) | 0.854 | 0.99 (0.22 to 4.37) | 0.989 |
| Other | 0.89 (0.20 to 3.90) | 0.879 | 1.10 (0.24 to 5.16) | 0.902 |
| Country polio status, n (%) | | | | |
| Non-endemic | Ref | Ref | Ref | Ref |
| Endemic | 0.97 (0.60 to 1.58) | 0.918 | 0.99 (0.58 to 1.67) | 0.96 |

*OR from the logistic regression model was adjusted for GPEI programme characteristics, organisational characteristics, external settings, years of surveillance experience, primary role and country polio status.
GPEI, Global Polio Eradication Initiative; Ref, Reference.

within communities was a major setback to the implementation of polio surveillance.[25] Similarly, in certain parts of the African region including South Sudan and Congo DRC, geographical barriers have led to logistical failures in transportation, which negatively impacted surveillance activities.[1] As such, even when the organisational setting is ideal, it is critical to account for external influences and conduct a routine evaluation of the entire surveillance programme to ensure that processes and strategies are tailored to surmount prevailing challenges. Our findings highlight the need for well-structured organisations to lead surveillance programmes. The organisation's work climate and culture need to be conducive to the learning, growth, productivity and innovative spirit of employees involved in surveillance activities.

We found that the more years of experience in surveillance activities the less likely were respondents to report implementation process as a factor for successful surveillance, regardless of the level of involvement. Our data showed that an individual's years of experience in polio surveillance activities becomes counterproductive between 15 and 19 years at the national level, during which success is less likely to be reported. This finding underscores the importance of continued improvement in surveillance processes and individual capacities over time. The dynamic nature of the processes for surveillance implementation requires that individuals consistently stay up to date with relevant skills needed to achieve desired objectives. It has been previously documented that active surveillance and education of surveillance officers were important in the rapid identification of cases and interruption of transmission during the Ebola outbreak of 2014.[12]

At the national level, challenges from organisational settings had an impact on the success being reported. This was different from what was seen at the district level where organisational challenges were noticed not to be a major facilitator to success. A disconnect between

national level (largely leaderships) and field-level staff, in terms of planning, expected outcomes, local external context and shortages could also account for the difference. In India, it has been reported that success was hinged on careful microplanning and strong operations at the national and state level, as well as accountability, social mobilisation and increased human resources at the district and subdistrict levels.[6] For global surveillance programmes, national-level leadership and coordination are very important for translating the goals of the global programme into country objectives and are likely to be more vulnerable to organisational dysfunction which can have ripple effects at the district levels.

This study is not without limitations. First, the study is prone to recall bias as participants reported their experiences over some time in the past. Also, possible misclassification due to the subjective definition of successful polio surveillance or polio challenges may have biased and weakened the validity of our results. In addition, given the cross-sectional study design, we are unable to infer causality from this study. There may also have been some residual confounding arising from peculiarities at the country level which may not have been fully accounted for in our analyses. Furthermore, we were unable to conduct subanalyses at the individual country level or for regions of Southeast Asia, Eastern Mediterranean and Western Hemisphere due to model instability from small cell counts. Despite these limitations, this study is novel in that it reports challenges and facilitators to polio surveillance from the perspectives of participants directly involved in polio surveillance activities since the GPEI. Also, given that the data were collected at the global level, the results may be readily generalisable across countries and settings.

## CONCLUSIONS

The process of conducting surveillance is a critical element for the success of GPEI surveillance activities. Overall, programme characteristics (eg, the perception that surveillance approaches are foreign to local communities and complex) may have impeded the process of conducting surveillance while organisational characteristics (eg, structural characteristics and organisational readiness) may have facilitated this process. These findings were slightly different at the national level and district level. Overall, the findings of this study are relevant for the planning and implementation of surveillance activities for the control of diseases including the current COVID-19 pandemic. Surveillance programmes should be designed with inputs from local communities and frontline implementers. Moreover, infectious disease surveillance initiatives will benefit from implementation by well-structured and well-resourced organisations to promptly manoeuvre around challenges.

**Contributors** IO conceived the paper, conducted the quantitative data analysis, and interpretation and coauthored the initial draft and subsequent drafts of the paper. PA contributed to data interpretation and authorship of subsequent drafts of the paper. AO conducted qualitative analysis and authorship of subsequent drafts of the paper. OA contributed to data interpretation, authorship of subsequent drafts of the paper, provided a review of the manuscript for intellectual content and was responsible for the overall content as guarantor. All authors read and approved the final manuscript.

**Funding** This study was funded by the Bill and Melinda Gates Foundation (Award Number: OPP1178578).

**Competing interests** None declared.

**Patient and public involvement** Patients and/or the public were not involved in the design, or conduct, or reporting, or dissemination plans of this research.

**Patient consent for publication** Not applicable.

**Ethics approval** The study was submitted to the Johns Hopkins Bloomberg School of Public Health Institutional Review Board and was deemed to be 'non-human subject research' (Approval Number: N/A). Participants gave informed consent to participate in the study before taking part.

**Provenance and peer review** Not commissioned; externally peer reviewed.

**Data availability statement** Data are available upon reasonable request. The data sets used and/or analysed during the current study are available from the corresponding author upon reasonable request.

**ORCID iDs**
Ikponmwosa Osaghae http://orcid.org/0000-0002-7202-7435
Priyanka Agrawal http://orcid.org/0000-0003-1314-7143

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
