## [Reviewer comments · BMJ Open]

ARTICLE DETAILS

TITLE (PROVISIONAL)	Facilitators and barriers of infectious diseases surveillance activities: lessons from the Global Polio Eradication Initiative, a mixed-methods study
AUTHORS	Osaghae, Ikponmwosa; Agrawal, priyanka; Olateju, Adetoun; Alonge, Olakunle

VERSION 1 – REVIEW

REVIEWER	Rumisha, Susan National Institute for Medical Research
REVIEW RETURNED	20-Feb-2022

GENERAL COMMENTS	Summary The paper contains useful information however, some sections are presented in a very complex manner hence lack clarity. It needs to be simplified. There is a circularity of description on what was studied with lots of mixed information. I suggest Authors to revisit this and make sure there is a consistence through out the manuscript. More specific comments: Objective and design There is a confusion on what is the objective of the study (how implementing factors influence the process?), due that it is defined differently and somehow complex at different sections of the paper – abstract, introduction, data analysis and conclusion. The analysis described looked at how the process affects/contributes to success – however, it seems like there is a 2-tier type of the research problem here, of which the descriptions are difficult to capture how this was taken into account. I suggest revising the objective to be simple, short and clear ... something like “... factors for a successful surveillance program” (which is what most of the results focus on). I struggled to understand if different analysis was done for each of the GPEI implementation factor? This is not mentioned in methods but results are presented for different group of individuals. Authors may use an illustration to describe the design of the study – this may bring clarity on how the outcome variable(s) were defined, explanatory variables and any categorizing of the individuals. Also may help the readers to understand how the process, characteristics and success are related structural-wise. Authors presented a tier of analysis which feeds into a another - but this is not very clearly described – needs to be revisited and simplified. Countries involved Would be useful to indicate when (which years or since when) GPEI
--

was implemented in these countries. If there is any variation, it should be mentioned and if possible included in the analysis (model) – as it may influence responses and effect of other factors. Also, authors should consider looking closely to countries where polio is still endemic vs. those which is eradicated – there may be lessons/challenges – the current paper does not provide any highlight on this.

Subset analysis

Authors conducted sub-group analyses at levels of implementation which is great.

Was there an attempt to conduct analysis at country level? only Africa which has the most of the individuals? If yes, can these be discussed? If not, could author explain why they think this may not be necessary? I would recommending looking at this, e.g., the over-representation of Africa and variations of running surveillance program in different countries may bias the observed findings and masking some effects.

Definitions

Success: Supplemental Table 1 should also include how “Success” was defined by respondents. Authors mentioned this to be “subjectively defined by respondents” of which I found it to be too vague since it is used to define the OUTCOME.

Dependent variable: this section requires either an illustration or a flow diagram. It is not so easy to understand.

Independent variables: Respondents characteristic are used here as explanatory variable; what is the difference between Individuals characteristics (used in the dependent variable) and the Respondents characteristics (used here)? Does this mean these characteristics were used in both? If not, then clarify this to avoid ambiguities.

Role of respondent should be included as a variable in the model: funder, manager, policymaker, ...frontline staff – lesson learns and opinions from these categories are crucial for other surveillance programs.

Considering the complex nature of the paper – authors should be consistence on how terms are defined and referred to – e.g., an outcome variable is called different in the “measures” and “data analysis” sub sections.

Results

The results section includes quotes from respondents – may be an intention of authors to retain the confidentiality here – however, a coding system could be used to make these findings informative – e.g., knowing the level of the respondent involvement, region, or role – views of a frontline worker may differ so much from that of a policymaker or a funder (see proposal to have this accounted in the analysis).

See other recommendations on subset analyses.

Discussion and conclusion

Paragraph 2 and 3 could be shortened – they include lots of information which not necessarily interpret findings from this study but rather explain how GPEI was run – I suggest author to move such information in the Introduction and restrict discussion on explaining what they found and what it means.

Some of the discussions could be supported by country specific or role-specific analysis (e.g. page 20, 21). Since these data are available -authors should attempt to analyse and use the findings to

	strengthen their arguments (i.e., not necessarily reporting in the main manuscript). Editorials Check name of countries - to all start with cap. letter Revisit Reference - formats
--	--

REVIEWER	Harvala, Heli NHS Blood and Transplant, National Microbiology Services
REVIEW RETURNED	23-Feb-2022

GENERAL COMMENTS	This study describes how implementation factors relating to the Global Polio Eradication Initiative (GPEI) program influenced the process of conducting surveillance at national and sub national level. The manuscript is difficult to follow, especially without a prior personal experience of GPEI programme, and hence I would like to suggest the following considerations. Study population included variety of individuals (i.e., funders, managers, policymakers, researchers, and frontline field implementers) from many different countries including seven LMICs. Did the authors consider how their background might have influenced the survey response? It would also be very useful to see how many countries and institutions were involved in this survey and were multiple participants from same country/institution allowed to join the study? Was there a particular reason why those 1,624 (44.38%) individuals who had been involved with GPEI activities from prior to 1988 were excluded from this study, assuming many of them would have continued in their role post 1988? I think it is important to consider what defines the success of the surveillance activities. Was the success of the surveillance activities defined by the participants only, or was that systematically measured based on pre-defined criteria? Results are difficult to follow and understand; please consider how to present them in a more generalised format. Include supplementary Table 2 as main Table. Would it be possible to include the actual questionnaire as Supplementary Material? Discussion is generally very interesting but at times present findings which have not been explained in results sections (such as: The expansion of surveillance focal sites beyond health facilities to include community surveillance was one of the internal contributors to successful surveillance in our study).
---

VERSION 1 – AUTHOR RESPONSE

Reviewer: 1

Dr. Susan Rumisha, National Institute for Medical Research

Comments to the Author:

Summary

The paper contains useful information however, some sections are presented in a very complex manner hence lack clarity. It needs to be simplified. There is a circularity of description on what was

studied with lots of mixed information. I suggest Authors to revisit this and make sure there is a consistence through out the manuscript.

Thank you for your review and comments. We have revised the entire manuscript to ensure the consistency of terminologies.

More specific comments:

Objective and design

There is a confusion on what is the objective of the study (how implementing factors influence the process?), due that it is defined differently and somehow complex at different sections of the paper – abstract, introduction, data analysis and conclusion. The analysis described looked at how the process affects/contributes to success – however, it seems like there is a 2-tier type of the research problem here, of which the descriptions are difficult to capture how this was taken into account. I suggest revising the objective to be simple, short and clear ... something like “... factors for a successful surveillance program” (which is what most of the results focus on).

I struggled to understand if different analysis was done for each of the GPEI implementation factor? This is not mentioned in methods but results are presented for different group of individuals.

Authors may use an illustration to describe the design of the study – this may bring clarity on how the outcome variable(s) were defined, explanatory variables and any categorizing of the individuals. Also, may help the readers to understand how the process, characteristics and success are related structural-wise.

Authors presented a tier of analysis which feeds into a another - but this is not very clearly described – needs to be revisited and simplified.

We have simplified the objective of the study to provide clarity and reflected this change in the abstract (page 2) and main manuscript (page 5).

Also, for clarity, we have included a flow chart as a supplemental figure to illustrate how the dependent and independent variables were conceptualized. Please see page 3 of the supplemental document for the flow chart (Supplemental Figure 1).

Each of the GPEI implementation challenges (organizational challenges, GPEI program challenges, and external challenges) were used as independent variables (see Figure 1 of supplemental document for flow chart) in each model. This has now been clarified with the aid of a flow chart and further described in the Methods section (pages 8-9). We also included a footnote for each of the tables.

Countries involved

Would be useful to indicate when (which years or since when) GPEI was implemented in these countries. If there is any variation, it should be mentioned and if possible, included in the analysis (model) – as it may influence responses and effect of other factors.

Our analytical sample was restricted to 1988 the year when GPEI was launched. This paper focuses on lessons learned from the GPEI since its launch in 1988 across all countries. As such, we did not include “time since the launch of GPEI” as a variable in our analysis. This has been further clarified at the beginning of our introduction (page 4) and in the analysis section on page 9.

Also, we are unable to include individual countries in our model because of model instability from small cell counts for some of the 70 countries represented in the analyses. We do not have information on the implementation intensity of various GPEI activities in each of these 70 countries

since the GPEI was launched in 1988.

Also, authors should consider looking closely to countries where polio is still endemic vs. those which is eradicated – there may be lessons/challenges – the current paper does not provide any highlight on this.

We have included a new variable “country polio status” in all models to help compare polio-endemic countries with countries where polio has been eradicated and this was included in all models (see pages 8-9, and 13). This has also been discussed briefly (page 19).

Subset analysis

Authors conducted sub-group analyses at levels of implementation which is great.

Was there an attempt to conduct analysis at country level? only Africa which has the most of the individuals? If yes, can these be discussed? If not, could author explain why they think this may not be necessary? I would recommending looking at this, e.g., the over-representation of Africa and variations of running surveillance program in different countries may bias the observed findings and masking some effects.

Thank you for this comment. We agree that a separate sub-analysis for the African region is of value. We have now included a new table 5 (Please see page 18) with a sub-analysis for the African region and discussed our findings (pages 20-21).

Unfortunately, we are unable to conduct a similar sub-analysis for individual countries or for the other WHO regions due to model instability from small cell counts. This has been included as a limitation to our study. Please see page 23.

Definitions

Success: Supplemental Table 1 should also include how “Success” was defined by respondents.

Authors mentioned this to be “subjectively defined by respondents” of which I found it to be too vague since it is used to define the OUTCOME.

The definition of “success” by the respondents has been clarified in the methods section and included in Supplemental Table 1. See page 1 of the supplemental document and page 7 of the main manuscript.

Dependent variable: this section requires either an illustration or a flow diagram. It is not so easy to understand.

We have included the definition of the dependent variable in the main manuscript and elaborated on this in the Supplemental Table 1. We have also included a flow chart as a supplemental figure to illustrate the definition of the dependent variable. Please see page 3 of the supplemental document for the flow chart (Supplemental Figure 1).

Independent variables: Respondent’s characteristic are used here as explanatory variable; what is the difference between Individuals characteristics (used in the dependent variable) and the Respondents characteristics (used here)? Does this mean these characteristics were used in both? If not, then clarify this to avoid ambiguities.

Individual characteristics used in the dependent variable are different from those used in the independent variable.

For the dependent variable, individual characteristics relate to internal factors contributing to success of surveillance program based on the CFIR framework (such as individual knowledge, self-efficacy, stage of change, perception, and other personal attributes). To avoid ambiguity, we have clarified this as “individual attributes” in Supplemental Table 1 and in the Methods section (page 7) and with the use of a flow chart (Supplemental Figure 1).

For the independent variable, respondents’ characteristics used are demographic characteristics relating to the implementation of the GPEI such as region, level of involvement, years of experience, and the primary role of implementing surveillance program.

We have included a flow chart as a supplemental figure to illustrate the definition of both our dependent and independent variables. Please see page 3 of the supplemental document for the flow chart (Supplemental Figure 1).

Role of respondent should be included as a variable in the model: funder, manager, policymaker, ...frontline staff – lesson learns and opinions from these categories are crucial for other surveillance programs.

Thank you for this comment. We agree and have included the primary role of respondents in surveillance activity in all models. Results have been reported and findings discussed. Please see pages 8-9, 13, and 19.

Considering the complex nature of the paper – authors should be consistency on how terms are defined and referred to – e.g., an outcome variable is called different in the “measures” and “data analysis” sub sections.

Inconsistencies in the definition of terms including the outcome variable have been addressed across the paper.

Results

The results section includes quotes from respondents – may be an intention of authors to retain the confidentiality here – however, a coding system could be used to make these findings informative – e.g., knowing the level of the respondent involvement, region, or role – views of a frontline worker may differ so much from that of a policymaker or a funder (see proposal to have this accounted in the analysis).

See other recommendations on subset analyses.

As rightly suggested, we have included primary role and country polio status – endemic vs non-endemic country as additional variables in all models to provide more context in interpretation of our results (please see pages 8-9, 13, and 19). Also, we have conducted additional subset analysis for the African region (pages 18, and 20-21).

Qualitative data from free-text responses were included in the results to validate findings from our quantitative analyses. Also, additional information such as the level of respondent’s involvement and role has been added to the qualitative results to provide additional information and context (please see pages 11-15). For example, on page 11 we stated, “According to a subnational surveillance officer from our qualitative analysis, process activities identified as factors for successful surveillance included “the setting up of community-based surveillance which involved community informants within their cells or villages, health workers and effective collaboration between community teams and district officials in disease detection, reporting, investigation, feedback, and outbreak response.”

Discussion and conclusion

Paragraph 2 and 3 could be shortened – they include lots of information which not necessarily interpret findings from this study but rather explain how GPEI was run – I suggest author to move such information in the Introduction and restrict discussion on explaining what they found and what it means.

We have shortened paragraphs 2 and 3 and made them more succinct and relevant to our results.
See page 19

Some of the discussions could be supported by country specific or role-specific analysis (e.g. page 20, 21). Since these data are available -authors should attempt to analyse and use the findings to strengthen their arguments (i.e., not necessarily reporting in the main manuscript).

We have included primary role and country polio status – endemic vs non-endemic country as additional variables in all models to provide more context in the interpretation of our results (please see pages 8-9, 13, and 19).

Editorials

Check name of countries - to all start with cap. letter

Revisit Reference - formats

We have edited the name of countries and formatted our references.

THANK YOU

Reviewer: 2

Dr. Heli Harvala, NHS Blood and Transplant

Comments to the Author:

This study describes how implementation factors relating to the Global Polio Eradication Initiative (GPEI) program influenced the process of conducting surveillance at national and sub national level. The manuscript is difficult to follow, especially without a prior personal experience of GPEI programme, and hence I would like to suggest the following considerations.

Study population included variety of individuals (i.e., funders, managers, policymakers, researchers, and frontline field implementers) from many different countries including seven LMICs. Did the authors consider how their background might have influenced the survey response? It would also be very useful to see how many countries and institutions were involved in this survey and were multiple participants from same country/institution allowed to join the study?

Thank you for your comments. We have now included the role of participants and their countries' polio endemicity status throughout the analyses to account for how the background of the participants influenced the implementation process of successful surveillance program. Please see pages 8-9, 13, and 19.

Also, we have clarified the number of countries identified by participants where polio surveillance program was implemented. A total of 70 countries were reported as countries of polio surveillance program implementation. Please see page 9.

Was there a particular reason why those 1,624 (44.38%) individuals who had been involved with GPEI activities from prior to 1988 were excluded from this study, assuming many of them would have

continued in their role post 1988?

There was no GPEI prior to 1988, though some individuals were involved in various polio vaccination activities in different countries. This study focused on lessons from the GPEI since the initiative was launched in 1988 – and all those who continued to implement surveillance activities under the GPEI after 1988 were included in our analyses. For clarity, we have included a flow chart (Supplemental Figure 1) of the study population on page 3 of supplemental document.

I think it is important to consider what defines the success of the surveillance activities. Was the success of the surveillance activities defined by the participants only, or was that systematically measured based on pre-defined criteria?

We have clarified how success was defined by including a definition of success in the methods section (page 7) and in Supplemental Table 1. “Success was defined as the respondent’s perceived accomplishment of the primary objective for implementing surveillance activities”.

Results are difficult to follow and understand; please consider how to present them in a more generalised format. Include supplementary Table 2 as main Table. Would it be possible to include the actual questionnaire as Supplementary Material?

We have edited the result section to ensure consistency and improve understanding.

We have also conducted an additional sub-analysis for the African region and included a new Table 5 as requested by reviewers (see page 18). We are unable to include Supplemental Table 2 as a main Table due to limits on the number of main tables and/or figures that can be included.

The study questionnaire is published (Alonge et al. 2020), and we have added the citation.

Discussion is generally very interesting but at times present findings which have not been explained in results sections (such as: The expansion of surveillance focal sites beyond health facilities to include community surveillance was one of the internal contributors to successful surveillance in our study).

This statement is a discussion of findings from the qualitative analysis presented in the result section. On page 10 we stated that “A district polio program supervisor described the “expansion of surveillance focal sites and adoption of new strategies like environmental surveillance” within communities.” For clarity, the entire discussion has been revised. Specifically, we have further edited and revised this in the discussion section. Please see page 20.

THANK YOU

VERSION 2 – REVIEW

REVIEWER	Rumisha, Susan National Institute for Medical Research
REVIEW RETURNED	28-Apr-2022
GENERAL COMMENTS	The paper has improved considerably. Congrats to the Authors.

	These comments may be considered for clarity: (May be this was missed during the previous revision) - Explain in the methodology, how you move from Bivariate to multivariable models - i.e., selection of which variables are included and which are dropped? A common practice involve setting a cut-off level of significance for the bivariate stage and drop any variable that do not meet that. This makes your multivariable models robust and not masking existing significance. As it is reported now, seems all variables are in both analyses (Bivariate and multivariable) Authors needs to list in the limitation that by not standardizing what is considered as a "Successful surveillance activity" could result into weak content validity. Editorials Correct where it reads multivariate to read multivariable. Note - these are two different models and yours is Multivariable. Correct P-value to p-value (i.e., small letter p)
--	--